

# avidaR: an R library to perform complex queries on an ontology-based database of digital organisms

Raúl Ortega and  Miguel Angel Fortuna

Estación Biológica de Doñana, Consejo Superior de Investigaciones Científicas, Seville, Spain

## ABSTRACT

Digital evolution is a branch of artificial life in which self-replicating computer programs—digital organisms—mutate and evolve within a user-defined computational environment. In spite of its value in biology, we still lack an up-to-date and comprehensive database on digital organisms resulting from evolution experiments. Therefore, we have developed an ontology-based semantic database—avidaDB—and an R package—avidaR—that provides users of the R programming language with an easy-to-use tool for performing complex queries without specific knowledge of SPARQL or RDF. avidaR can be used to do research on robustness, evolvability, complexity, phenotypic plasticity, gene regulatory networks, and genomic architecture by retrieving the genomes, phenotypes, and transcriptomes of more than a million digital organisms available on avidaDB. avidaR is already accepted on CRAN (*i.e.*, a comprehensive collection of R packages contributed by the R community) and will make biologists better equipped to embrace the field of digital evolution.

## INTRODUCTION

Over the past 30 years, digital evolution research has established itself as a valuable approach in biology, bridging experimental research with computational modelling (*Ray, 1991*). The contribution of digital evolution to the development of ecology and evolutionary biology comprises diverse topics such as robustness and evolvability (*Edlund & Adami, 2004*; *Lenski, Barrick & Ofria, 2006*; *Elena et al., 2007*; *Elena & Sanjuán, 2008*; *Fortuna et al., 2017*), complexity (*Ray, 1997*; *Lenski et al., 1999*; *Adami, Ofria & Collier, 2000*; *Gerlee & Lundh, 2008*; *Ofria, Huang & Torng, 2008*), phenotypic plasticity (*Clune, Ofria & Pennock, 2007*; *Lalejini et al., 2021*; *Fortuna, 2022*), the role historical contingency in evolution (*Hagstrom et al., 2004*; *Wagenaar & Adami, 2004*; *Clune et al., 2012*),  ecological interactions among species (*Cooper & Ofria, 2003*; *Johnson & Wilke, 2004*; *Zaman, Devangam & Ofria, 2011*; *Fortuna et al., 2013*; *Zaman et al., 2014*; *Dolson & Ofria, 2021*), gene regulatory networks (*Lenski et al., 2003*; *Edlund & Adami, 2004*; *Covert et al., 2013*), genomic architecture (*Wilke et al., 2001*; *Knibbe et al., 2005*; *Adami, 2006*; *Knibbe et al., 2007*; *Gerlee & Lundh, 2008*; *Batut et al., 2013*; *Gupta et al., 2016*), evolution of sex (*Chandler, Ofria & Dworkin, 2012*),

Corresponding author
Miguel Angel Fortuna,
fortuna@ebd.csic.es

and evolution of cooperation (*Goings et al., 2004*; *Knoester, McKinley & Ofria, 2007*; *Clune et al., 2011*), among others.

Avida is the most widely-used software platform for the study of evolution (*Ofria & Wilke, 2004*). Within this computational framework of open-ended evolution, digital organisms, which are self-replicating computer programs, engage in competition for limited resources such as memory space and central processing unit (CPU) time. The genome of a digital organism consists of a circular sequence of code instructions that is continuously executed by a virtual CPU. Certain instructions within the genome facilitate the replication of the organism, acting as the sole mechanism for transmitting its genetic material to future generations. However, this replication process is susceptible to errors or mutations. When an instruction is inaccurately copied, it is replaced in the offspring genome by a randomly selected instruction from a set of 26 possibilities, distributed uniformly. Other instruction sets are available, for example when working with parasites (*i.e.,* organisms that steal CPU cycles from other organisms—their hosts—to speed up their own replication process). The resulting variation generates competition among organisms as the population grows and uses up available resources. In order to metabolize these resources and thus replicate, organisms must compute Boolean logic operations—such as *NOT* and *NAND*—on binary numbers taken from the environment through input–output instructions encoded in their genomes. The organism that is fittest, in the sense of being the faster replicator while encoding the phenotype required to metabolize a resource in a specific environment, comes to dominate the population through natural selection.

In spite of the extended used of Avida, no effort has been made to build and provide access within the scientific community to an up-to-date and comprehensive database on digital organisms. We have developed the avidaR package to provide users of the R programming language (*i.e.,* an open-source statistical programming language widely used for data analysis, statistical modeling, and graphical visualization) with easy-to-use tools for performing complex queries on avidaDB—an ontology-based database containing the genomes, phenotypes and transcriptomes of more than a million digital organisms obtained from evolution experiments carried out in Avida.

## DIGITAL EVOLUTION

### avidaDB

We have built a comprehensive database that encompasses the genomes, phenotypes, and transcriptomes of over one million digital organisms resulting from evolution experiments carried out using Avida. A significant portion of the data, more than half, was obtained by randomly sampling the sequence space of genomes that were 100 instructions long (*Fortuna et al., 2017*). These genomes encoded the same phenotype within a single environment. Furthermore, in a subsequent study (*Fortuna, 2022*), the phenotype encoded by the genomes of these 512,000 organisms, as well as their transcriptomes, were computed in a set of 1000 distinct environments. The database includes the following key data from these two experiments:

- genome instruction sequence: a linear string of letters representing the instruction codes that make up the genome of a digital organism.
- phenotype: a unique combination of logic operations that a digital organism performs on 32-bit one- and two-binary input numbers.
- transcriptome instruction sequence: a linear string of letters representing the instruction codes that make up the transcriptome executed by a digital organism.

In a third study (Rico and Fortuna, unpublished), we extended the analysis by computing the phenotype and transcriptome of all single-point mutants for 200 digital organisms from the previous dataset. These single-point mutants were generated by introducing a single mutation at different positions in the genome. The computations were performed in the same 1,000 environments as in the previous study. As a result, we obtained an additional dataset comprising 500,000 digital organisms (2500 single-point mutants for each of the 200 organisms). In this expanded dataset, we have stored the same type of data as in the previous studies, including genomes, phenotypes, and transcriptomes. Moreover, we have also included information about the specific genomic locations of all single-point mutations:

- mutant of: a relation linking two digital organisms if their genomes differ in a single instruction code.
- position at genome: the position of an instruction code along the digital genome of a digital organism.

In addition to the genomic, phenotypic, and transcriptomic data, we have also computed and stored information on the viability and generation time of the 1,012,000 digital organisms in the same 1,000 environments:

- viable: the ability of a digital organism to produce an offspring able to replicate by executing its genome.
- generation time: number of instruction codes executed by a digital organism to produce a viable offspring (*i.e.,* the inverse of fitness in the absence of any selective pressure other than reducing the number of instructions that must be executed for successful replication).

## Tracing the provenance and reproducibility of data

Clarifying the provenance of data is crucial for traceability and reproducibility in scientific research. We have added functions into avidaR to trace the context and provenance of the data obtained from Avida experiments:

- get_experiment_id_from_organism_id() # establishing a connection between a digital organism and the experiment it was derived from enables researchers to easily retrieve the relevant experiment information associated with that specific organism.
- get_doi_from_experiment_id() # establishing a connection between an experiment and the identifier of the scientific publication it was reported in enables researchers to easily access the publication, which provides additional details on the experimental design.

- get_docker_image_from_experiment_id() # establishing a connection between and experiment and a docker image that contains the configuration files and options used in the experiment enables researchers to reproduce the experimental conditions and obtain repeatable results.

## Providing semantics to avidaDB

avidaDB has been implemented as an RDF store (also known as graph database or triple store) to express relationships between resources using subject–predicate–object triples. The most relevant advantage of RDF stores is that they represent, store and query data as a graph. The second characteristic is that they are semantic, which means that they can store not only data but also explicit descriptions of the meaning of that data. These explicit descriptions are known by the RDF and linked data community as ontologies. An ontology is a machine-readable description of a domain that typically includes a vocabulary of terms and some specification of how these terms relate to each other, imposing a structure on the data for such domain. This is also known as a schema.

We have developed the Ontology for Avida (OntoAvida) to provide semantics to avidaDB. The semantic relationships between the terms commonly used in Avida are expressed in the W3C standard ontology language OWL-DL. OntoAvida is part of the Open Biological and Biomedical Ontologies (https://obofoundry.org/ontology/ontoavida.html) and is already available to the scientific community. We have also customized pyLODE to obtain a readable version of OntoAvida, which is also available at https://owl.fortunalab.org/ontoavida.

## Connecting to avidaDB

Semantic databases are databases that store RDF data and allow the querying of RDF data *via* the SPARQL query language (*i.e.*, a programming language used to retrieve and manipulate data stored as subject–predicate–object triples). The library avidaR can connect to triple-stores that support the RDF4J server REST API such as GraphDB. The SPARQL endpoint of avidaDB (https://graphdb.fortunalab.org/) allows both basic connection (requiring no password or requiring basic HTTP user-password authentication) or connection secured with an API access token. The way avidaR connects avidaDB requires user-password authentication (*i.e.*, public_avida as both username and password).

avidaR retrieves data from avidaDB without specific knowledge of SPARQL or RDF. The functions in avidaR can be grouped by the type of data requested: (i) genomes as sequences of letters representing code instructions (*e.g.*, get genomes encoding specific phenotypes in particular environments), (ii) phenotypes as a combination of the logic operations computed (*e.g.*, get the phenotype of organisms that computed specific logic operations), (iii) transcriptomes executed during the replication cycle as both sequences of letters representing code instructions and chord diagrams (*e.g.*, get the transcriptomes of organisms whose genomes encoded specific phenotypes in specific environments), and (iv) single-point mutants at specific positions on the genome (*e.g.*, get the genome of the single-point mutant that substituted the instruction coded by letter i by j at a particular position on the genome of a wild-type organism). Users can build programming workflows

by piping multiple functions together so that the output of one becomes the input to another (*e.g.*, get a genome encoding different phenotypes in distinct environments (*i.e.,* plastic organism), and retrieve the generation time as a surrogate of fitness in the absence of any selective pressure other than reducing the number of instructions executed).

## R FUNCTIONS

Below are some of the functionalities that can be found in avidaDB:

### avidaDB access (required to establish a connection to the database)
- avidaDB <- triplestore_access$new() # create triplestore object.
- avidaDB$set_access_options(url = "https://graphdb.fortunalab.org", user = "public_avida", password = "public_avida", repository = "avidaDB") # set access options.

### avidaDB summary
- get_db_summary(triplestore = avidaDB) # get a summary of the content of the database.

### Genetic language used by the organisms in Avida
- instruction_set(inst_set = "heads") # get the default Avida genetic language.

### Logic operations computed by digital organisms
- logic_operation() # get the list of logic operations that a digital organism can compute (*i.e.,* the default logic-9 environment).
- get_logic_operation_from_phenotype_id() # get the list of logic operations computed by a digital organism whose genome encodes a specific phenotype out of the 512 distinct phenotypes comprising the entire phenotype space (*i.e.,* ranging from phenotype 0 to phenotype 511).

### Genomes harbored by digital organisms
- get_genome_id_from_logic_operation() # get genome identifiers of organisms that computed specific logic operations.
- get_genome_id_from_phenotype_id() # get genome identifiers encoding specific phenotypes in particular environments.
- get_genome_id_from_transcriptome_id() # get genome identifiers of organisms that executed specific transcriptomes in particular environments.
- get_genome_id_from_genome_seq() # get genome identifiers of genome sequences.
- get_genome_seq_from_genome_id() # get genome sequence from genome identifiers.

### Phenotypes encoded by genomes of digital organisms
- get_phenotype_id_from_genome_id() # get the phenotype identifier encoded by a genome identifier.
- get_phenotype_id_from_genome_seq() # get the phenotype identifier encoded by a genome sequence.
- get_phenotype_id_from_logic_operation() # get the phenotype identifier of organisms that computed specific logic operations.

- get_phenotype_id_from_transcriptome_id() # get the phenotype identifier of organisms that executed specific transcriptomes in specific environments.

### Transcriptomes executed by digital organisms

- get_transcriptome_id_from_genome_id() # get the transcriptome identifier encoded by a genome identifier.
- get_transcriptome_id_from_genome_seq() # get the transcriptome identifier encoded by a genome sequence.
- get_transcriptome_id_from_logic_operation() # get the transcriptome identifier of organisms that computed specific logic operations.
- get_transcriptome_id_from_phenotype_id() # get the transcriptome identifier of organisms whose genomes encoded specific phenotypes in specific environments.
- get_transcriptome_seq_from_transcriptome_id() # get the transcriptome sequence from the transcriptome identifier.
- plot_transcriptome(inst_set = "heads") = # plot a transcriptome as a chord diagram using the default instruction set.

### Single-point mutants

- get_genome_id_of_wild_type_organisms() # get the genome identifier of the list of organisms for which genomic, transcriptomic, and phenotypic information of all their single-point mutants is available (*i.e.,* wild-type organisms).
- get_mutant_at_pos() # get the genome sequence of specific single-point mutants at a particular position on the genome of a wild-type organism.

## CASE STUDIES

We illustrate the use of avidaR by addressing two case studies: (i) quantifying the fitness cost of phenotypic plasticity, and (ii) identifying mutations with positive effects on replication time as a surrogate of fitness. The analysis was performed using avidaR version 1.2.0.

### The fitness cost of phenotypic plasticity

In the first case study, we investigate the impact of phenotypic plasticity on the fitness of digital organisms. Phenotypic plasticity is not only a pervasive feature of biological organisms but also of digital ones (*Fortuna, 2022*). Specifically, the environment can modify the sequence of instructions executed from a digital organism's genome (*i.e.,* its transcriptome), which results in changes in its phenotype (*i.e.,* the ability of the digital organism to perform Boolean logic operations). This epigenetic pathway for plasticity comes at a fitness cost to an organism's viability and replication time: the longer the replication time (higher fitness cost), the more chances for the environment to modify the genetic execution flow control, and the higher the likelihood for the genome to encode novel phenotypes.

We obtain a set of genomes that exhibit phenotypic plasticity, where organisms can switch between different phenotypes in response to changes in their environment. We also collect a set of genomes that do not exhibit plasticity and maintain a fixed phenotype

across all environments. By comparing the replication times of organisms with and without plasticity in different environments, we quantify the fitness cost associated with phenotypic plasticity. Here is the code to retrieve the data and perform the analysis.

```r
# load required libraries:
library(avidaR)
library(tidyverse)

# data reproducibility:
set.seed(1000)

# create triplestore object:
avidaDB <- triplestore_access$new()

# set access options:
avidaDB$set_access_options(
url = "https://graphdb.fortunalab.org",
user = "public_avida",
password = "public_avida",
repository = "avidaDB"
)

# define function to get the data:
data <- function(boolean_func, seeds, plastic) {
    get_genome_id_from_logic_operation(
        # logic operations that the target organisms must perform:
        logic_operation = boolean_func,
        # number of distinct environments where that phenotype will be computed:
        seed_id = 1:seeds,
        triplestore = avidaDB) %>%
    # show data for each genome that computes those logic operations:
    group_by(genome_id) %>%
    # number of environments where the phenotype was encoded:
    summarize(n_seeds = n()) %>%
    # classify the genome as plastic or no plastic:
    filter(if (plastic == TRUE) n_seeds == seeds else n_seeds == 1) %>%
    # randomly select 10 organisms from those that meet the requirements:
    sample_n(10) %>%
    select(genome_id) %>%
    # get the transcriptome from the genome of each selected organisms:
    get_transcriptome_id_from_genome_id(
        genome_id = as.integer(gsub("genome_", "", .$genome_id)),
        transcriptome_seq = TRUE,
        seed_id = 1:seeds,
        triplestore = avidaDB) %>%
    select(genome_id, transcriptome_id, transcriptome_seq) %>%
    distinct %>%
    # get the number of instructions that the organism executed to replicate as a surrogate for fitness:
    mutate(generation_time = nchar(transcriptome_seq),
        phen_group = boolean_func, plastic_group = plastic) %>%
    select(-transcriptome_seq)
```

```
}

# get the data and join them in a single dataframe:
# select 10 non-plastic organisms performing the logic function NOT:
data_NOT_non_plastic <-
    data(boolean_func = "not", seeds = 10, plastic = FALSE)
# select 10 plastic organisms performing the logic function NOT:
data_NOT_plastic <-
        data(boolean_func = "not", seeds = 10, plastic = TRUE)
# select 10 non-plastic organisms performing the logic function EQUALS:
data_EQU_non_plastic <-
        data(boolean_func = "equals", seeds = 10, plastic = FALSE)
# select 10 plastic organisms performing the logic function EQUALS:
data_EQU_plastic <-
        data(boolean_func = "equals", seeds = 10, plastic = TRUE)
# provide data as a single data frame:
df <- rbind(
    data_NOT_non_plastic, data_NOT_plastic,
    data_EQU_non_plastic, data_EQU_plastic)

# plot the results:
ggplot(df, aes(x = phen_group, y = generation_time, fill = plastic_group)) +
geom_boxplot()
```

This preliminary finding suggests that plasticity may incur a fitness cost for simple phenotypes, but not for complex phenotypes. This is evidenced by an increased time for digital organisms to replicate, indicating a decrease in fitness (Fig. 1).

## The fitness effects of single-point mutations

In the second case study, we retrieve data from avidaDB to identify the single-point mutation that has the largest positive effect on replication rate. We consider the replication rate as a surrogate for fitness, where a higher replication rate corresponds to higher fitness. Replication in digital organisms involves the copying of the genome instruction by instruction into a new memory region, a process that can occasionally result in errors known as mutations. Mutations occur when an instruction is inaccurately copied, leading to its replacement in the offspring genome with a randomly selected instruction. Mutations can have different effects on fitness, including being lethal (resulting in non-viable offspring), neutral (no change in phenotype or fitness), deleterious (decreasing fitness), or beneficial (increasing fitness).

The process of identifying the most significant single-point mutation on replication rate involves multiple steps:

- randomly sampling the genome sequence of a single organism (wild-type organism).
- retrieving the transcriptome of the wild-type organism and its single-point mutants in a specific environment.
- selecting the mutant with the highest fitness based on generation time as a surrogate measure.

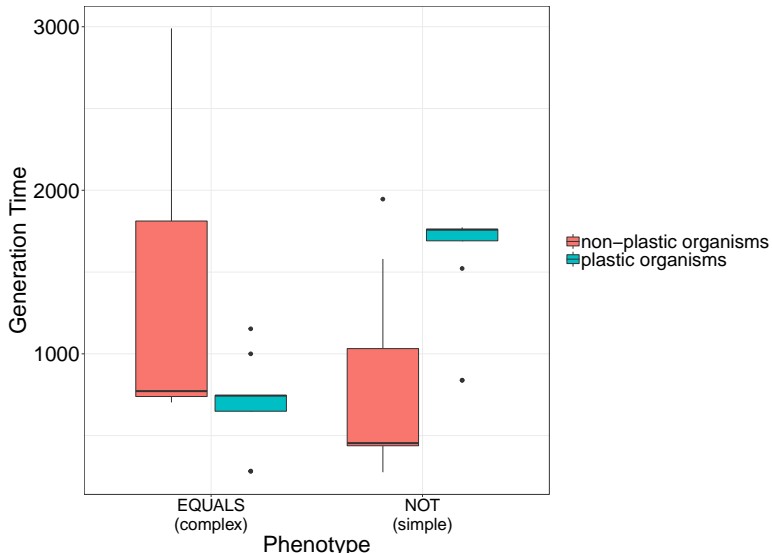

**Figure 1** **Simple analysis to illustrate the use of avidaR to quantify the fitness costs of phenotypic plasticity.** We suggest that plastic organisms whose genomes encode a simple phenotype (*i.e.*, digital organisms able to compute the Boolean operation NOT) have longer generation times (*i.e.*, lower fitness) than non-plastic ones. In contrast, organisms whose genomes encode complex phenotypes (*i.e.*, digital organisms able to compute the Boolean operation EQUALS) show the opposite. Generation time was measured as the number of instructions that a digital organism must execute to produce an offspring.

- comparing the genome sequences of the wild-type organism and the fittest mutant to identify the specific mutated instruction and its position in the genome.
- finally, plotting the transcriptomes of the wild-type organism and the fittest mutant for comparison (Fig. 2).

Here is the code to retrieve the data and perform the analysis.

```
# load required libraries:
library(avidaR)
library(tidyverse)

# data reproducibility:
set.seed(10)

# create triplestore object:
avidaDB <- triplestore_access$new()

# set access options:
avidaDB$set_access_options(
url = "https://graphdb.fortunalab.org",
user = "public_avida",
password = "public_avida",
repository = "avidaDB"
)
```

```
# get data:
data_wild_type <-
    get_genome_id_of_wild_type_organisms(
        triplestore = avidaDB) %>%
    # randomly select the genome of an organism for which all of its single-point mutants are known:
    sample_n(1) %>%
    # get the transcriptome from the genome of the selected organism:
    get_transcriptome_id_from_genome_id(
        genome_id = as.integer(gsub("genome_", "", .$genome_id_wild_type)),
        transcriptome_seq = TRUE,
        seed_id = 1,
        triplestore = avidaDB) %>%
    # get the number of instructions that the organism executed to replicate as a surrogate for fitness:
    mutate(generation_time = nchar(transcriptome_seq)) %>%
    select(genome_id, transcriptome_id, generation_time) %>%
    arrange(generation_time)
# get all single-point mutants from the selected organism:
data_mutant <-
    # specificy the position of the mutation along the genome of the organism:
    get_mutant_at_pos(
        genome_id = as.integer(
            gsub("genome_", "", data_wild_type$genome_id)),
        triplestore = avidaDB) %>%
    select(genome_id_mutant) %>%
    # get the transcriptome from the genome of the mutant organisms:
    get_transcriptome_id_from_genome_id(
        genome_id = as.integer(gsub("genome_", "", .$genome_id_mutant)),
        transcriptome_seq = TRUE,
        seed_id = 1,
        triplestore = avidaDB) %>%
    select(genome_id, transcriptome_id, transcriptome_seq) %>%
    distinct %>%
    # get the number of instructions that the mutant executed to replicate as a surrogate for fitness:
    mutate(generation_time = nchar(transcriptome_seq)) %>%
    select(-transcriptome_seq) %>%
    # sort the mutants by the number of instructions that the mutant executed to replicate:
    arrange(generation_time) %>%
    # select a single mutant organism, i.e., the one with the highest fitness:
    head(1)

# join data for both organisms in a single dataframe:
df <- inner_join(
    rbind(data_wild_type, data_mutant),
    get_genome_seq_from_genome_id(
        genome_id = c(as.integer(
            gsub("genome_", "", data_wild_type$genome_id)),
                    as.integer(
            gsub("genome_", "", data_mutant$genome_id))),
        triplestore = avidaDB),
    by = "genome_id")
```

```
# get the relative fitness of the mutant compared to that of the wildtype:
df$generation_time[1] / df$generation_time[2]

# get the position of the mutation in the genome of the mutant:
pos <- mapply(function(x, y) which(x != y)[1],
    strsplit(df$genome_seq[1], ""), strsplit(df$genome_seq[2], ""))

# get the mutated instruction:
substr(df$genome_seq[1], pos, pos)
substr(df$genome_seq[2], pos, pos)

# plot the transcriptomes of both organisms:
plot_transcriptome(
    inst_set = "heads",
    transcriptome_id = as.integer(
        gsub("transcriptome_", "", df$transcriptome_id[1])),
    seed_id = 1,
    save = TRUE, save_path = getwd(), format = "pdf",
    triplestore = avidaDB)
```

A single-point mutation has the potential to significantly impact fitness by altering the execution flow of the instructions within a digital organism's genome. In our analysis, we observed a notable fitness improvement resulting from a specific mutation. Specifically, the substitution of the math instruction *n* (dec) at position 54 in the genome with the control-flow instruction h (jump-head) led to a 29% increase in the replication rate of the digital organism (Fig. 2). This finding highlights the important influence that individual mutations can have on the performance and fitness of digital organisms.

## FUTURE DIRECTIONS

Digital evolution presents a highly promising avenue within the field of ecology and evolution, offering an intermediary level of complexity between real-life systems and traditional mathematical models. By studying digital organisms, researchers gain un-precedented insights into evolutionary processes that are otherwise challenging to explore using natural systems alone. Consequently, digital organisms offer a complementary approach to studying natural or experimental evolution. It is important to note, however, that while predictions from evolving digital organisms can provide valuable insights, there may exist substantial differences in the underlying mechanisms between digital and biological organisms. Nonetheless, the fundamental operational processes, such as Darwinian evolution, remain equivalent. Thus, studies using digital organisms can help us elucidate the principles governing evolutionary dynamics across various scales.

The avidaR library was developed as part of a larger project aimed at providing semantics to the data generated from experiments carried out on the Avida software platform. It represents the first R package that enables users to perform advanced queries on an ontology-based semantic database, which stores the genomes, phenotypes, and

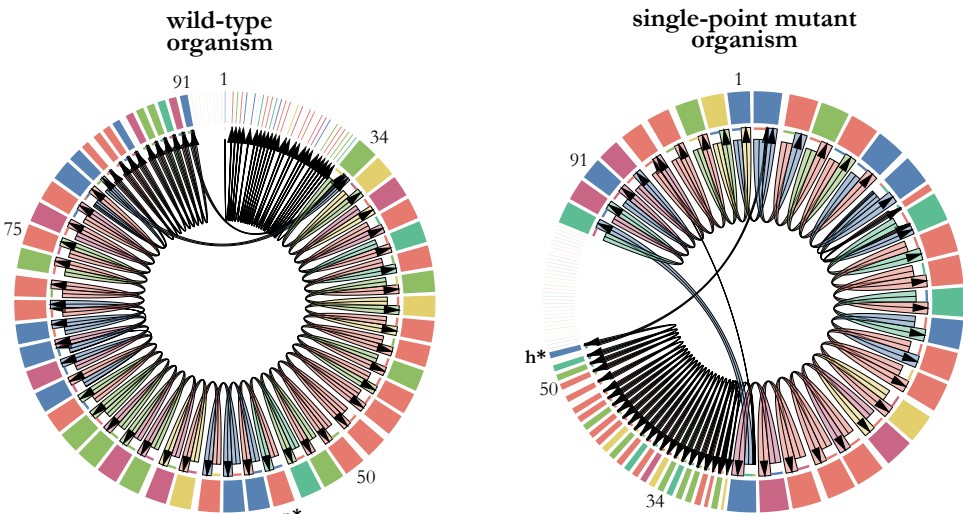

**wild-type organism**

**single-point mutant organism**

**Figure 2** **Changes in transcription as a consequence of a single-point mutation.** The transcriptomes of a wild-type digital organism (left) and its single-point mutant with the largest positive effect on fitness (right) are represented as chord diagrams. The instructions of the 100-length genome that were actually executed are represented as segments around the circle. The size of each segment is proportional to the number of times that the instruction was executed, and its color depicts the type of instruction. Gaps between segments indicate parts of the genome that were not executed. The first instruction of the genome that is executed is placed at the top center of the chord diagram. Some numbers indicating genome positions were placed along the genome to illustrate the changes in the execution flow that took place when the math-instruction labelled as *n* at position 54 was substituted by the control-flow instruction labelled as h. This single-point mutation, depicted in bold and marked with an asterisk, increases 29% the replication time of the mutant organism.

transcriptomes of over a million digital organisms resulting from evolution experiments. This integrated framework empowers biologists by equipping them with the necessary tools to explore and analyze the field of digital evolution more effectively. By leveraging the functionalities of avidaR, researchers can gain deeper insights into the evolutionary processes and dynamics of digital organisms. The power of avidaR as a tool for evolutionary analysis is exemplified by two case studies that highlight its capabilities in addressing complex questions related to fitness, plasticity, and genetic variation. The Jupyter Notebook available at the following URL https://gitlab.com/fortunalab/avidaR/-/blob/main/avidaR.ipynb, provides additional examples of the versatile applications of avidaR. In this notebook, you will find detailed examples and code snippets showcasing various features and functionalities of avidaR. It serves as a comprehensive resource for users to explore and understand the capabilities of the package. By working through the provided examples, researchers can gain practical insights into how to effectively utilize avidaR for their own evolutionary analyses and experiments.

We will expand avidaDB by including the genomes of parasite organisms from our previous and ongoing research. Parasitic digital organisms are almost identical to any other organism, and as such they compute Boolean logic operations and self-replicate by copying their genome instruction-by-instruction into a new memory space. But they

operate inside hosts, stealing CPU cycles from them to execute their own genome's instructions and, hence, reduce their host fitness.

avidaR is a valuable tool to perform studies that lie at the intersection of evolutionary biology and computer science. Its potential transcends traditional academic boundaries and will definitely provide fertile ground for new collaborations bridging these disciplines. Our R package will appeal not only to students of evolutionary biology and computer science, but also to synthetic biologists, systems engineers, students of the origin of life, and philosophers.

## ACKNOWLEDGEMENTS

We would like to thank the Bioinformatics and Computational Biology (BCB) lab (bcb.ebd.csic.es) at the Doñana Biological Station (EBD-CSIC), for the computational support and services provided.

### Funding
This work was supported by the Spanish Ministry of Science and Innovation through the Ramon y Cajal Program (RyC2018-024115-1) and the Knowledge Generation Grant Program (PID2019-104345GA-I00), as well as by the Plan Andaluz de Investigacion, Desarrollo e Innovacion (PAIDI 2020) of Junta de Andalucía (PY20_00765). The funders had no role in study design, data collection and analysis, decision to publish, or preparation of the manuscript.

### Grant Disclosures
The following grant information was disclosed by the authors:
Spanish Ministry of Science and Innovation through the Ramon y Cajal Program: RyC2018-024115-1.
Knowledge Generation Grant Program: PID2019-104345GA-I00.
Plan Andaluz de Investigacion, Desarrollo e Innovacion (PAIDI 2020) of Junta de Andalucía: PY20_00765.

### Competing Interests
The authors declare there are no competing interests.

### Author Contributions
- Raúl Ortega performed the experiments, analyzed the data, performed the computation work, prepared figures and/or tables, and approved the final draft.
- Miguel Angel Fortuna conceived and designed the experiments, performed the experiments, analyzed the data, prepared figures and/or tables, authored or reviewed drafts of the article, and approved the final draft.

## Data Availability

avidaR is available at GitLab and at Zenodo:

- https://gitlab.com/fortunalab/avidaR.

- Ortega, Raúl, & Fortuna, Miguel A. (2023). avidaR: an R library to perform complex queries on an ontology-based database of digital organisms. https://doi.org/10.5281/zenodo.8094788.

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
