# Peer review of "avidaR: an R library to perform complex queries on an ontology-based database of digital organisms"

_PeerJ Computer Science, doi:10.7717/peerj-cs.1568_

## Round 0.1 · original submission · Major Revisions

Please revised the paper accordingly. Then it will be evaluated again.

Reviewer 1 ·

Basic reporting

This manuscript includes clear, unambiguous, professional English
language. I have no revision suggestions on language, grammar, and text flow.

Introduction and background need additional information on the subject. I noticed that Supplementary material contains further information on this area of interest (paragraph 1. Digital Evolution). I suggest that you add this information into the body of manuscript, resulting in the more profound Introduction section. In addition, you should consider removing most of the data/information, including figures, tables, main text, from Supplementary material into the manuscript file. Only technical details can be left in the Supplementary file. This is how readers can find all the necessary information in one file, not jumping between Supplementary and manuscript files. After doing this, you may remove the following sentence from Abstract: “A detailed tutorial and additional case studies are provided in the Supplementary Material.”, because information is now included in the manuscript.

About the literature and references in Introduction. This paper/research article is included in the field of computer science and many readers are familiar with the subject, but I suggest that you introduce some terms and concepts, such as R programming, SPARQL, RDF, and CRAN. Only short descriptions or introductions will do. In addition, you mention an “Author comment” on the PeerJ guidance page. This is an important and relevant thing. I suggest that you add this information in your manuscript (Introduction/background section), because it includes basic facts about the material and methods you used in this work.

I understand that this paper is included in the PeerJ Computer Science section of “Data Handling and Mining”, but I still think that this manuscript should be structured in the form of a scientific paper. Please also check the journal guidelines if it is needed to add the List of Abbreviations at the end of the manuscript, since multiple abbreviations appear in this paper.

Figures are relevant, high quality, well labelled, and described. However, they are not discussed in the main text of this paper. More discussion about the figures should be added, and how the findings differ from prior work/literature.

Raw data and code are supplied and checked.

Experimental design

I think that the research question should be defined more clearly, although the aim/objective/purpose of this study can be seen in text. This research is relevant and meaningful. However, this study is not filling the knowledge gap properly right now, thus should be identified more closely (Please see my comment above). More information is needed to answer the question why it is important to study this subject.

Validity of the findings

More information on the impact, novelty, rationale, and benefits of this research needs to be added in order to show the readers why it is important to study this area of research. This can be done by embedding more literature with prior research findings. Additional information should be embedded in Abstract and also in Introduction. My suggestions are described above.

All underlying data have been provided. They are robust and statistically sound.

Conclusion: This section needs more information. You should discuss more about how scientists/biologists can benefit your results. What are the areas where and how the results can be used? In addition, future work related to your results should be discussed, meaning that what kind of further research is going to be done or should be done in the near future.

The findings of your work should be discussed more profoundly, including the comparison to prior work with relevant references/literature added in text and the Reference section. Some limitations related to your work may also appear, thus should be discussed.

Additional comments

This manuscript should be re-structured and further information added in order to meet the criteria of a scientific paper. Please see above. This paper should be re-reviewed after revisions.

·

Basic reporting

This manuscript overviews the avidaR package, which is a new R package developed to interface with a database containing the genomes, phenotypes, and transcriptomes of digital organisms from the Avida Digital Evolution Platform.

The manuscript is well-written, easy to follow, and generally provides sufficient context for readers.

Areas of improvement are given below. I see the first two as critical revisions.

1. Given that the primary purpose of avidaR is to interface with avidaDB, the main manuscript needs more background on avidaDB. For example, how were digital organisms represented in the database generated? This information appears to be in the supplemental material (the "avidaDB" section). The majority of this supplemental section should be integrated into the main manuscript.

2. Context is essential for interpreting the genetic / phenotype information from Avida output files (and by extension, the information stored in avidaDB). For the most part, it appears that the data accessed by avidaR were generated using default Avida settings. This contextual information should be clarified in the manuscript (and ideally in the avidaR documentation).

3. Given the introductory nature of the two case study code snippets, I suggest adding more explanatory code comments or annotations throughout.

Experimental design

No comment (N/A).

Validity of the findings

No comment (N/A).

Additional comments

I installed the avidaR package on my own machine (R version 4.2.1, avidaR version 1.1.3) and verified the case study code. The graphs produced were not identical to those in the paper; however, the differences were largely cosmetic.

Overall, avidaR is a valuable contribution, lowering the barrier to entry to digital evolution research and addressing a very real need in digital evolution research. Beyond being a useful resource for research, I see this as a valuable tool for incorporating digital evolution into classroom activities.

Prior to publication, I think the main article needs to provide additional context on avidaDB (as detailed in previous comments).

Suggestions:

1. Line 9: Some in the field of evolutionary computing would disagree, as digital evolution experiments generally aren't focused on problem-solving. I suggest removing the connection to "evolutionary computation", but I'll leave that to the authors' discretion.

2. Include package version information for your case studies.

3. Not necessary, but it would be useful to discuss your plans for additions to avidaDB. Is there an open call for contributions? Do you plan to add more genomes in the future?

4. Related to the previous suggestion, it could also be useful to discuss possible extensions to the avidaR package (if any).

Typos

Line 13: "...with an easy-to-use tools…" → "...with easy-to-use tools…"

Line 47, 48: Same problem as on line 13.

Line 143: "Finaly" → "Finally"

·

Basic reporting

The manuscript of Ortega & Fortuna introduces a database of Avidian digital organisms (avidaDB) as well as the R package avidaR to help researchers interface with this database. Digital evolution is a very promising direction within ecology and evolution, as it represents an intermediate step between the complexity of real-life systems and that of traditional mathematical models. Having a tool to make its use easier is a very valuable contribution. Below I give some suggestions for improving the presentation, and on some future possibilities for extending the avidaR package.

Perhaps the most important point is that the avidaR package, at this time, is mostly tailored to interacting with the avidaDB database. While that is perfectly fine and perfectly useful, I think that a large part of the reason why many theoretical biologists are reluctant to adopt Avida as a research tool in the first place is that the cost of entry is perceived to be a bit too high to be worth it. Putting it more directly: many biologists might feel that it is not worth investing precious time into learning assembly language and weird systems of syntax for manipulating Avida environments and analyzing digital organisms - at least not for the expected gains. This is exactly where a package like avidaR could, in time, become something that helps people by lowering the barrier to entry. Many, many utility functions to perform standard Avida tasks could be added to make life easier. I am thinking of functions for setting up Avida config files and digital experiments, analyzing Avida detail files, manipulating environments, performing phylogenetic analyses, creating more types of plots, and so on. Are there plans to extend avidaR in these directions? If so, then I would allude to this in the text.

A second, much more technical, but perhaps important point: the avidaR package has functionalities for converting single-letter abbreviated instructions into full assembly code, or showing what assembly instructions are available (e.g., convert_seq_into_org, instruction_set, ...). But Avida allows for several different instruction sets. For example, when running instruction_set(), there are no instructions for injecting parasites and the like; only the basic instruction set is shown. I assume that many organisms in the avidaDB database are not products of this standard instruction set. Is there an option for specifying which instruction set should be used in, e.g., convert_seq_into_org? This would not just be useful, but (as far as I understand) absolutely necessary for interpreting the digital organisms correctly.

Third, some standard Avida logic operations have pre-set names. For example, the first argument of get_genome_id_from_logic_operation is a logic operation. According to the help page, this can be one of "equals", "exclusive-or", "not-or", "and-not", "or", "orn-not", "and", "not-and", and "not". But since Avida has at least 80 operations that organisms can perform to gain resources from the environment, this list feels quite short. Would it be possible to extend this parameter to receive all valid Avida operations?

Finally, something that is not terribly important nor possible to implement without an API-breaking change, but I am mentioning it nevertheless. It is the issue of the names of the functions in the package. They are very long, and often unnecessarily so. In particular, prefacing functions with "get_" is not a great idea. Out of the 28 functions listed in the index help page, 23 start with "get_". This does not increase clarity ("tandem_id_from_genome_id" is just as expressive as "get_tandem_id_from_genome_id"), but makes functions more clunky. Could a terser convention be adopted? Such as, say, "genome2phenotype" instead of "get_phenotype_id_from_genome_id"? I will not belabour this point, but I did want to at least signal that there is room for improving user experience here.


Additionally, I have the following more specific comments:

Line 47: I was happy to see that avidaR is already on CRAN - it would be good to provide the link here.

Line 48: tools -> tool

Line 48: What is an "ontology-based" database?

Lines 87-133 and 145-211: These code snippets could be included as vignettes in avidaR itself, or put on some public repository for download. The point is that the reader shouldn't have to copy-paste code from a pdf (I did that, and all my quotes turned into equality signs, for instance). I would also typeset code with monospace font and apply syntax highlighting. R Markdown or Quarto Markdown will be useful tools for doing this.

Lines 99, 109, 155, 161, 169, 175, 191, 211: In each of these places, the code refers to the object 'avidaDB', which is undefined. I had to look up the help page of get_genome_id_of_wild_type_organisms to see how one can fix this problem. The supplied scripts could therefore start with the following:

avidaDB <- triplestore_access$new()
avidaDB$set_access_options(
url = "https://graphdb.fortunalab.org",
user = "public_avida",
password = "public_avida",
repository = "avidaDB_test"
)


Lines 135-136, 138: Perhaps a minor nuance, but ecologists certainly won't be happy to see "fitness" being equated with "speed of replication". I think the Authors know exactly why. To remediate, I would simply make a side comment that in Avida, "fitness" refers to speed of replication, which need not be indicative of who would win in competition with other phenotypes.

Line 137, 140, and Figure 2: single-point mutation -> single point mutation


Sincerely,
Gyuri Barabás

Experimental design

N/A

Validity of the findings

N/A

Additional comments

N/A

---

## Round 0.2 · Minor Revisions

This version has satisfied all reviewers. However, one reviewer has some minor suggestions. Please revise it accordingly. Then the paper is potentially acceptable.

Reviewer 1 ·

Basic reporting

I read this manuscript throughout. Authors have made revisions on this paper. However, I still have some minor technical comments listed below:

-Introduction: "CPU" should be written out when mentioned for the first time in text.
-2.2 Tracing the provenance and reproducibility of data: ...associated with that specific organism." Is this quotation mark (") at the end of this sentence a typo? I see no other quotation mark. Please clarify and revise.
-Case studies: "(see Fig. 1)" and "(see Fig. 2)" You may simply say "(Fig. 1)" and "(Fig. 2)" as said in the latter paragraph "(Fig. 2)".
-Future directions: "...in the Avida software platform." Please consider using "on" preposition.

Experimental design

No comment.

Validity of the findings

No comment.

Additional comments

This manuscript has definitely improved. I suggest that this paper is accepted.

·

Basic reporting

The authors' revisions are well-integrated and sufficiently address the concerns outlined in my initial review.

Experimental design

no comment

Validity of the findings

no comment

Additional comments

The authors have addressed the comments in my initial review. I recommend that the article be accepted.

·

Basic reporting

no comment

Experimental design

no comment

Validity of the findings

no comment

Additional comments

I thank the Authors for the thorough and careful job they have done with the revisions. I have no further comments - except to say that I believe the Authors' work will be a very useful contribution to both ecology and evolutionary biology.

Sincerely,
Gyuri Barabás

P.S.: While I must confess I am still not sure how "get_phenotype_id_from_genome_id" increases clarity compared with either "phenotype_id_from_genome_id" or even just "genome2phenotype", I will also keep my promise and avoid pouncing on this (in any event, very minor) point.

---

## Round 0.3 · accepted · Accept

Thank the authors for their efforts to improve the work. I believe this version is ready for acceptance. Congrats!